# Slice-level Detection of Intracranial Hemorrhage on CT Using Deep Descriptors of Adjacent Slices

## 1. Introduction

In many tasks of medical image analysis, the high-resolution 3D volumes of Computed Tomography (CT) scans pose formidable computational challenges for training deep networks. We propose in this paper a new strategy to train *slice-level* classifiers on CT based on the descriptors of the adjacent slices along the axis, each of which is extracted through a convolutional neural network (CNN). This method is applicable to CT datasets with per-slice labels such as the RSNA Intracranial Hemorrhage (ICH) dataset (RSNA, 2019), which aims to predict the presence of ICH and classify it into 5 different sub-types.

The proposed method exploits a two-stage training scheme. In the first stage, we treat a CT scan simply as a set of 2D images and train a state-of-the-art CNN classifier (He et al., 2016) that was pretrained on ImageNet. During the training, each slice is sampled *together* with the 3 slices before and the 3 slices after it, which makes the batch size a multiple of 7. In the second stage, the output *descriptors* of each block of 7 consecutive slices obtained from stage 1 are stacked into an image and fed to another CNN for final prediction of the middle slice. Our model is entirely trained on the RSNA dataset and additionally evaluated on the CQ500 dataset (Chilamkurthy et al., 2018), which adopts the same a set of labels but only on study level. We obtain a single model[1] in the top 4% best-performing solutions of the RSNA ICH challenge, where model ensembles are allowed. Experiments also show that the proposed method significantly outperforms the baseline model on CQ500.

## 2. Proposed approach

It is difficult to exploit high-performing deep neural networks for the ICH classification while keeping the full 3D resolution of the input CT. On the other hand, using slices as independent 2D images and ignoring the axial information of data causes detrimental effects to the algorithm's performance. Our method gets the best of both worlds: we apply transfer learning (Pan and Yang, 2009) on 2D images to perform classification per slice and then assemble the results of local slices to refine the prediction of the middle one. The proposed scheme, illustrated in Figure 1, consists of two stages: descriptor extraction and axial fusion.

### 2.1. Descriptor extraction

In this stage, we train a 2D CNN to classify individual slices of the CT scans, which are converted into 3-channel images using 3 different windows: brain ($l = 40, w = 80$), subdural

---

1. Model weights and codes are available upon acceptance of the paper.

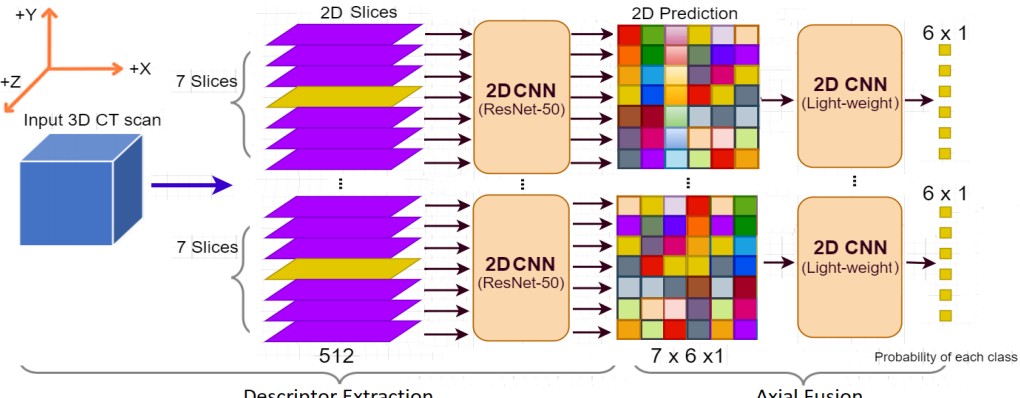

Figure 1: Illustration of the proposed two-stage training procedure.

$(l = 75, w = 215)$, and bone $(l = 600, w = 2800)$. The output of the network is a descriptor of size $6 \times 1$ that includes the probabilities of the 5 ICH sub-types and an additional class for any of them. During training, each slice is always sampled in a block of 7 that includes itself in the center and 6 neighboring slices. With this approach, we can take advantage of pre-trained models on ImageNet (Deng et al., 2009) to initialize the network. Specifically, a ResNet-50 (He et al., 2016) was used in our experiments. We followed the procedure in (He et al., 2019) and trained the network for 20 epochs with a batch size of $16 \times 7$ using Adam optimizer (Kingma and Ba, 2015). An initial learning rate of $5e - 4$ and the cosine annealing learning rate scheduler (Loshchilov and Hutter, 2017) were used. Several augmentation techniques such as cropping, resizing, flips, rotations, distortions, gaussian noise, and CutMix (Yun et al., 2019) were applied to prevent the network from overfitting.

## 2.2. Axial fusion

This stage combines the descriptors of each 7 consecutive slices generated in stage 1 to exploit the axial information and to refine the prediction of the centered slice. In particular, we concatenate the 7 descriptors into a $7 \times 6 \times 1$ tensor and train a 3-layer CNN to output the final classification result for the representative slice in the middle of the block. This network contains only 2 convolution layers and 1 fully connected layer. The 2D convolution kernels help the model learn both the relationship between ICH predictions across local slices and the relationship between probabilities for the sub-types. The output of the fusion network can therefore be seen as a re-calibrated prediction of a single slice.

## 3. Experiments and results

### 3.1. Datasets and evaluation protocols

The RSNA and CQ500 datasets were used to verify the effectiveness of the proposed approach. Both of them contain non-contrast CT scans that are labeled with 5 sub-types of ICH: *intraparenchymal, intraventricular, subdural, extradural,* and *subarachnoid.* The only difference between the two datasets is that the labels of RSNA are per slice, while the those of CQ500 are per CT scan. The whole RSNA dataset was split into 3 parts: a public training set (19,530 studies), a public testing set (2,214 studies), and a private testing set

(3,518 studies). Each study is a CT scan of 20 to 60 slices of $512 \times 512$ pixels. The weighted log loss was used as the evaluation metric for this dataset, in which a weight of 2/7 was used for the ICH label and a weight of 1/7 was used for each of the 5 sub-types.

Meanwhile, CQ500 consists of 500 studies, from which 490 were selected for experiments while the rest 10 of them are noisy and were excluded from the dataset. We used CQ500 as another test set to validate the efficiency and robustness of the proposed algorithm, which was merely trained on the public training set of RSNA. On this test, the performance of our method is measured by area under the ROC curve (AUC). Note that the study-level probability of any class was taken as the maximum probability amongst all slices.

### 3.2. Results

We report a weighted log loss of 0.05341 on the private testing set of RSNA, which ranks in top 4% over 1345 teams on the Kaggle leaderboard. Note that our result is provided by a single ResNet-50 model, while many other solutions in this competition exploit ensemble techniques. On CQ500, the proposed method achieves a mean AUC of 0.971. This is an improvement of around 2% compared to the baseline model (Chilamkurthy et al., 2018). Especially, our method provides better AUC scores over all disease labels as shown in Table 1. These results strongly demonstrate the generalization capacity of our model, which was trained on a different dataset with a different labeling protocol.

| Findings | (Chilamkurthy et al., 2018) | Ours |
|---|---|---|
| ICH (any subtypes) | 0.9419 | **0.9612** |
| Intraparenchymal | 0.9544 | **0.9691** |
| Intraventricular | 0.9310 | **0.9832** |
| Subarachnoid | 0.9574 | **0.9596** |
| Subdural | 0.9521 | **0.9694** |
| Extradural | 0.9731 | **0.9814** |
| Mean | 0.9520 | **0.9710** |

Table 1: Experimental results measured by AUC score on CQ500 dataset.

### 4. Conclusion

We presented in this paper a novel two-stage training strategy for the task of slice-level classification on CT scans. The key idea is to sample each slice together with its neighbors and to refine the classification of it using the coarse descriptors of the whole group. This method was experimentally demonstrated to work well on ICH datasets like RSNA and CQ500. We believe that it can be easily extended to other 3D datasets of CT or MRI scans.

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
