# OpenReview forum: "Slice-level Detection of Intracranial Hemorrhage on CT  Using Deep Descriptors of Adjacent Slices"
_MIDL.io/2020/Conference — Submitted to MIDL 2020_

### Official Review · AnonReviewer1 · 2020-03-12
**Slice-level detection of ICH on CT using a two-stage CNN that uses adjacent slices**

**Rating:** 1
**Confidence:** 5

**Review:**

In this paper, the authors propose a two-stage deep learning method for producing slice-wise preditions for intracranial hemorrhage detetion. The paper is well written and easy to follow. The authors first train a 2D convolutional network to perform classification per slice using transfer learning. During this training process, the 6 adjacent slices are part of the same batch. Subsequently, a second CNN takes the predictions for each class for the 7 slices to produce a 7x6x1 tensor which is used to produce an updated prediction for the center slice. The method is validated on the RSNA Hemorrhage challenge and the CQ500 dataset. A score of 0.05341 on the RSNA challenge would have ranked 41 in the challenge.

Pro's:
- Trained and validated on a large set. Also validated on an external dataset.
- Compared a publicly available benchmark.

Cons:
- I think this paper has little novelty.
- A logical comparison would be to compare this approach with pseudo 3D approaches where multiple input slices are fed as extra channels. This is a common method which adds little computational overhead. A comparison with this baseline would have been very relevant.
- A disadvantage of this approach is that it is not trained end-to-end. Why is this not possible? I do not see why not, and it is not explained in the paper.  The second stage could also be added using 1x1x7 convolutions? Also, the authors use batches of 16x7 so I would expect that it would fit in memory if the batch size is reduced.
- It would have been good to report the performance without the second stage to see how much that adds to the performance.
- From the paper, I understand that the authors only trained with the public training set of the RSNA challenge. Furthermore, performance is measured on the private test set. So, the public testing set was not used at all?

---

### Official Review · AnonReviewer4 · 2020-03-13
**The method is clear and the result shows considerable improvement, but lack of novelty.**

**Rating:** 3
**Confidence:** 3

**Review:**

This paper proposed a two-stage method, which first sample each slice and their neighbor for a coarse per-slice classification, then another network is used to refine the classification of the central slice using the output descriptors of the whole group.
1. The result shows around 2% increase in AUC and improves all predicted labels.
2. Method description is clear.
3. Novelty is only to add a refinement 3-layer CNN for the output of the first step, which is not quite enough.
4. The second step uses CNN instead of ANN, so that only "neighbor" labels' relationship is considered. It would be better to show the comparison to ANN in the ablation study.

---

### Official Review · AnonReviewer3 · 2020-03-14
**Slice level description using adjacent slices**

**Rating:** 3
**Confidence:** 4

**Review:**

This manuscript describes a two-stage training scheme. The first stage uses  2D slices of CT scans to train a CNN classifier. The second stage stacks a block of 7 consecutive slices from stage 1 and trains another CNN for final prediction. The proposed method was demonstrated on RSNA ICH dataset and also CQ500 dataset, and achieved good results.

The paper overall is well written. The method is not novel but has been demonstrated with good results. All the details of the algorithm have been clearly described.

---

### Official Review · AnonReviewer2 · 2020-03-16
**A clearly presented idea that lacks validation**

**Rating:** 1
**Confidence:** 4

**Review:**

This is short paper that targets integration of adjacent slices into the learning process for intra-cranial hemorrhage classification. The approach uses a multi-slice (slab) network followed by a single plane fusion network. Results on the RSNA challenge are reasonable (top 4%), but no direct numeric comparison is made with the leading methods. The method performs reasonably versus a baseline network on a private dataset.

The method is simple and easy to follow. The approach could be readily combined with other technologies. The results perform well against the chosen baseline method. The conclusion is well supported by the data.

The paper does not make a direct comparison against the RNSA leaderboard show that the technique would augment the highest ranked non-ensemble method from the board.

The level of novelty is not clear. Multiple groups have used multi-slice slab learning followed by fusion. It is not cleat that the proposed approach would outperform similar approaches already found in the literature.

No assessment of variance was performed. No statistical assessment / modeling was performed.

The limited assessment of variability and lack of sensitivity / ablation experiments renders the generalizability of the work difficult to understand in context. The lack of a leading comparison algorithms reduces confidence in the strength of the results.

---

### Meta-Review · Area_Chair1 · 2020-04-07
**MetaReview of Paper143 by AreaChair1**

**Rating:** 1

**Metareview:**

The work is generally seen to not have much technical novelty and also to exhibit significant limitations such as the lack of validation and missing baseline comparisons. This makes it difficult to recommend acceptance.

**Paper Type:**

validation/application paper

---

### Decision · Program_Chairs · 2020-04-11

Reject